# Therapy Defining at Initial Diagnosis of Primary Brain Tumor—The Role of ^18^F-FET PET/CT and MRI

**DOI:** 10.3390/biomedicines11010128

**Published:** 2023-01-04

**Authors:** Dávid Gergő Nagy, Imre Fedorcsák, Attila György Bagó, Georgina Gáti, János Martos, Péter Szabó, Hajnalka Rajnai, István Kenessey, Katalin Borbély

**Affiliations:** 1National Institute of Mental Health, Neurology and Neurosurgery, 1145 Budapest, Hungary; 2Scanomed Ltd., 1145 Budapest, Hungary; 3Department of Pathology and Experimental Cancer Research, Semmelweis University, 1085 Budapest, Hungary; 4National Cancer Registry, National Institute of Oncology, 1122 Budapest, Hungary; 5Pathology, Forensic and Insurance Medicine, Semmelweis University, 1091 Budapest, Hungary; 6PET/CT Outpatient Department, National Institute of Oncology, 1122 Budapest, Hungary

**Keywords:** primary malignant brain tumors, glioblastoma, amino acid tracers, ^18^F-FET PET/CT, non enhancing tumors, MRI, perfusion weighted MRI, drug treatment, neurological disorder

## Abstract

Primary malignant brain tumors are heterogeneous and infrequent neoplasms. Their classification, therapeutic regimen and prognosis have undergone significant development requiring the innovation of an imaging diagnostic. The performance of enhanced magnetic resonance imaging depends on blood–brain barrier function. Several studies have demonstrated the advantages of static and dynamic amino acid PET/CT providing accurate metabolic status in the neurooncological setting. The aim of our single-center retrospective study was to test the primary diagnostic role of amino acid PET/CT compared to enhanced MRI. Emphasis was placed on cases prior to intervention, therefore, a certain natural bias was inevitable. In our analysis for newly found brain tumors ^18^F-FET PET/CT outperformed contrast MRI and PWI in terms of sensitivity and negative predictive value (100% vs. 52.9% and 36.36%; 100% vs. 38.46% and 41.67%), in terms of positive predictive value their performance was roughly the same (84.21 % vs. 90% and 100%), whereas regarding specificity contrast MRI and PWI were superior (40% vs. 83.33% and 100%). Based on these results the superiority of ^18^F-FET PET/CT seems to present incremental value during the initial diagnosis. In the case of non-enhancing tumors, it should always be suggested as a therapy-determining test.

## 1. Introduction

Primary malignant brain tumors consist mainly of gliomas WHO grade 3 and 4. The most frequent entity is glioblastoma. Their annual incidence according to the estimates of GLOBOCAN is approx. 7.5 cases per 100,000 individuals in the US and 9.8 in the EU, while 7.9 in Hungary [1]. What makes them a huge burden for healthcare professionals is their dismal prognosis and enormous impact on quality of life. Despite all diagnostic and therapeutic efforts, the mean overall survival for glioblastoma is 15–20 months [2,3]. However, in certain subgroups, longer survival is observed [4]. Due to the evolution of genetic diagnosis, these subgroups have been more accurately characterized. WHO also incorporated these developments, since CNS taxonomy was updated twice over the last decade, putting more and more emphasis on genetic features [5]. For instance, molecular parameters (TERT promoter mutation, EGFR amplification, and +7/−10 copy number changes) in IDH-wildtype diffuse astrocytomas can justify a grade 4 histology [5]. Recent advances in diagnostics provide early diagnosis and treatment options for a wider range of patients. Currently, the first line treatment for grade 4 gliomas is surgical removal with maximal extent, if possible with adjuvant radiochemotherapy followed by temozolomide chemotherapy [6]. In accordance with the CATNON trial, adjuvant temozolomide therapy is advised for grade 3 astrocytomas after radiotherapy [7]. In the case of grade 3 oligodendrogliomas, adjuvant radiotherapy and PCV or TMZ chemotherapy are advised by EANO and NCCN guidelines. Increasing clinical experience has helped to identify subgroups most susceptible to these therapies, which can benefit the most from adequate timely therapy [4].

Parallel to these developments presurgical diagnostic imaging tools have also been going through a major evolution. Advanced MRI techniques like DWI, PWI and MRS were developed. Based upon these measurements cellular density, neovascularization and metabolite concentration can be calculated, implying the local metabolism and even IDH status providing more exact information about the state of the lesion [8]. However, these sequences still rely heavily on the blood–brain barrier function and also face the challenge of long-known non-enhancing malignant tumors (up to 40% of anaplastic gliomas) [9,10]. For accurate metabolic imaging, PET/CT has spread widely as an extremely valuable diagnostic tool in the case of systematic oncological diseases. Due to the high cortical glucose metabolism, the interpretation of ^18^F-FDG PET/CT data is often considered difficult in the detection of central nervous system tumors. However, since the development of suitable amino acid radiotracers [11,12,13], PET/CT has become ascendant in the neuro-oncological setting as well.

Several in vitro and in vivo studies provided data that amino acid uptake is independent of blood-brain barrier function, yet still presenting high tumor-brain contrast, the most precise tumor delineation and high tumor sensitivity. Regarding diagnostic performance and practical concerns, ^18^F-FET (O-(2-[18F]fluoroethyl)-L-tyrosine) became the most widely used radiotracer throughout Europe [14,15,16].

Comparing its diagnostic performance to enhanced MRI, numerous studies showed that elevated amino acid uptake detects non-enhancing tumor parts offering better biological tumor delineation (Figure 1 and Figure 2) [17,18,19,20]. This enables adequate surgical intervention, correct sampling and subsequent oncological treatment. Due to it being independent of blood-brain barrier integrity, treatment-related changes do not compromise the image quality throughout oncological treatment. One of the few true neurosurgical positive prognostic factors for glioma and glioblastoma survival is gross total resection [21]. The proper assessment of the real biological extent of the tumor and its relation to neural elements provides the possibility of advanced surgical planning.

The question of MRI-based supramarginal resection of gliomas has always been a controversial topic [22,23,24]. As amino acid PET/CT provides more accurate information about tumor extent, FET-PET- and 5-ALA-guided resection trials begin to determine its positive effect on survival. Preliminary results show that biological tumor volume on post-operative PET/CT may be one of the strongest prognostic factors for survival [25,26]. Another challenging topic of neurooncology was assessing treatment-related changes versus actual recurrence or residual viable tumor. In recent years, scientific research provided enough evidence for EANO/RANO guidelines to list amino acid PET/CT as an elemental part of follow-up diagnostics [27,28,29]. 

Due to the growing number of publications about ^18^F-FET PET/CT performance in treatment follow-up, the focus has shifted from its fundamental role in newly diagnosed tumors. Our goal was to assess the advantage of using amino acid PET/CT at the initial diagnosis and recommend exact indications for metabolic imaging. In our single-center retrospective study, we reviewed the diagnostic performance of ^18^F-FET/PET CT compared to that of contrast-enhanced and perfusion-weighted MRI imaging. 

## 2. Materials and Methods

As a routine examination, we have been using ^18^F-FET PET/CT since November 2019. In our on-site scanner (GE Discovery MI 4 (2017 edition) we were able to perform 197 examinations in 166 patients between November 2019 and October 2021. The referring physicians were all neurosurgeons, mostly with additional clinical oncologist licensing. Indications for amino acid PET scan were brain lesions of uncertain origin on MRI either at initial diagnosis or during oncological treatment. The average time difference between the scans was 30 days (1–109 day spectrum) with a median value of 22 days, with a slight contribution of the pandemic and the subsequent tracer supply problems. 

^18^F-FET was synthesized at the Radiopharmacological Department of the Nuclear Medicine Center, University of Debrecen Clinical Center, Hungary. Patients fasted 4 h prior to the intravenous administration of 200 MBq ^18^F-FET with 20-min uptake, followed by 10-min acquisition. Patients underwent non-contrast, low-dose CT and a subsequent PET scan at the same supine position; for image reconstruction, the OSEM and BPL methods were used. The maximum of the standardized uptake values (SUVmax) of the region of interest and contralateral normal tissue were measured by using InterView™ FUSION 3.08 (MEDISO Ltd., Budapest, Hungary) multi-modality medical image visualization and post-processing software. Subsequently, the maximal tumor-to-background ratio (TBRmax = SUVmax target/SUVmax background) was calculated. Evaluation of the PET scans was always done by two nuclear medicine specialists. MRI scans were previously assessed by specialists in neuroradiology and neurosurgery. Every lesion with a TBRmax over 1.6 was considered to be ^18^F-FET PET-positive and possibly a malignant lesion [29]. During our study, both the PET/CT and MRI scans were blindly re-evaluated by a neurosurgeon and either a nuclear medicine or a neuroradiology specialist. In terms of MRI evaluation, only visual analysis was performed. Adhering to our interrater agreement for statistical purposes, signs of contrast enhancement or perfusion elevation were handled as possibly malignant lesions.

All PET scans without previously known pathology were followed by surgical sampling. For follow-up cases, consideration of surgical accessibility and patient benefit was taken into account. The main intention was to test the diagnostic performance of ^18^F-FET PET at initial diagnosis. In order to avoid such selection bias, we reviewed every histologically verified case. The final number of analyzed scans was 30. In one case, the readings of ^18^F-FET data suggested a non-tumorous, inflammatory lesion, which was also histologically confirmed as a demyelinating lesion part of reactive tissue. Due to its non-neoplastic nature, we excluded that case from our further statistical analysis.

The average time difference between PET scan and surgical intervention was 37 days with a median value of 21 days and values varying from 1 to 142 days. Histological analysis was performed in the same center (Department of Pathology and Experimental Cancer Research, Semmelweis University, Budapest, Hungary) according to the current WHO CNS classification recommendation (5th edition). For histological features, hematoxylin-eosin stained sections were made, additional immunohistochemical staining (GFAP, IDH1 R132H, ATRX Ki-67) was prepared, if necessary, and genetic sequencing (1p19q co-deletion, IDH1/2 mutation) was also conducted. After histological confirmation, we compared the radiological and pathological results. Grade 2 histology was treated clinically as a low-grade lesion. Grade 3 and 4 histological samples were handled together as clinically malignant tumors requiring adjuvant therapy.

We compared the performance of diagnostic modalities in accordance with the actual histological diagnosis. In order to test the screening ability of the aforementioned examinations sensitivity, specificity, positive predictive value and negative predictive value were calculated. We evaluated the correlation between SUVmax, TBRmax, Ki-67 (a nuclear protein associated with cell proliferation) and tumor grading with Student’s t-test or the Mann-Whitney test, depending on the distribution of samples. Grade and Ki-67 proliferation index were determined by a neuropathologist expert (H.R., MD, Ph.D.) on H&E-stained slides and immunohistochemistry, respectively. Ki-67 proliferation index was determined as part of a routine, fully automated (Leica Bond-Max) pathological protocol at the Department of Pathology and Experimental Cancer Research, Semmelweis University, Budapest, Hungary. The Ki-67 proliferation index (percentage of positive cells) was determined using the anti–Ki-67 antibody (Dako clone: MIB1 #M7240). In addition, the associations between SUVmax or TBRmax with Ki-67 were assessed using Spearman’s rank order correlation. Fisher’s exact test was utilized to analyze the relationship between IDH status and either SUVmax and TBRmax or contrast enhancement and perfusion. *p*-values under 0.05 were accepted as statistically significant. Statistical analyses were performed by Microsoft Excel for Mac 16.56 (Microsoft, Redmond, WA, USA) and Statistica 14.01.25 (TIBCO Software, Stanford Research Park, Palo Alto, CA, USA).

## 3. Results

The cohort contained 29 brain tumor patients. Male: female ratio was 16:13 (Table 1). The median age was 42 years (range: 26–79 years). The location of the scanned lesions was predominantly in the frontal lobe (62.1%) followed by parietal (17.2%), insular (13.8%), temporal (3.4%) and occipital (3.4%) regions, combined with a slight right dominance (58.6–41.4%). 

### 3.1. PET/CT

Twenty-six scans were ^18^F-FET PET positive, with a mean and median TBRmax of 2.7 and 2.45, respectively. From the three negative scans in one case, we registered slightly elevated (SUVmax: 1.8 vs. 1.6 TBRmax: 1.1) and largely identical (SUVmax: 1.23 vs. 1.21 TBRmax: 1.02) amino acid uptake compared to the healthy contralateral white matter. There was one photopenic examination with a TBRmax of 0.81 (SUVmax: 1.5 vs. 1.8; Figure 3). Based on recommendations from the literature this was handled as a distinct subgroup and not included in either false negative or true positive categories [30,31,32]. 

Twenty-three of the twenty-six positive cases proved to be histologically also malignant, meaning an 88.46% positive predictive value. In our cohort, there were no false negative scans, meaning a 100% negative predictive value and sensitivity. Because of three false-positive scans, which were all WHO grade 2 gliomas, the specificity was 40%. 

From the false-positive scans in one case during clinical follow-up, we noticed an unexpectedly early (14 months after surgery, 8 months after radiotherapy) and multifocal progression from the patient—though a follow-up ^18^F-FET PET scan 4 months before the progressive MRI finding was also a negative value. The one ^18^F-FET avid tumor was operated on in an elective fashion due to the previous misinterpretation of the state of the lesion, meaning the time difference between isotope scan and surgery was 142 days. A follow-up MRI examination (120 days after PET) showed a slight contrast enhancement in the cranial part of the tumor as a sign of progression.

### 3.2. CE and rCBV MRI

All patients underwent contrast-enhanced MRI, but unfortunately only in 22 cases were additional perfusion sequences calculated. Comparing the histological result with previous MRI findings, we found that, in terms of contrast enhancement, sensitivity was 58.33%, specificity 83.33%, positive predictive value 93.3% and negative predictive value 33.33%. For perfusion-weighted imaging, sensitivity was 33.33%, specificity 100%, positive predictive value 100% and negative predictive value 29.41%.

Looking at these parameters, only the newly diagnosed tumor cases by ^18^F-FET PET have largely identical values, and PPV drops to only 84.21%. Sensitivity and positive predictive value of contrast enhancement decrease (52.9% and 90%), specificity does not change and NPV increases minimally (38.46%). Perfusion-weighted imaging also performs slightly better, with both sensitivity and NPV showing a subtle increase (36.36% and 41.67%), while PPV and specificity remain the same (Table 2).

While low-grade tumors did not show contrast enhancement, 41.7% of the high-grade group featured this parameter, which comparison proved to be significant (Table 3). Elevated rCBV was absent in the low-grade group, however, the relative difference in frequency was not statistically significant. Among the high-grade subgroup, statistically higher TBRmax (*p* = 0.043) and Ki-67 (*p* = 0.004) were evaluated (Figure 4).

Correlation analysis revealed a mild association between SUVmax and Ki-67, however, the latter proved to be independent of TBRmax (Figure 5). 

Regarding the genetic status of the lesions, we did not find any correspondence between IDH status and TBRmax (Figure 6), contrast enhancement, or perfusion (Table 4). 

## 4. Discussion

Adequate imaging diagnostics is one of the biggest challenges of neuro-oncology. Both anatomical and metabolic relations should be well visualized irrespective of surgical or oncological treatment and timing of examination [33,34]. Enhanced MRI sequences have been the gold standard modality for the last two decades. The results of our study support and agree with previous literature data that contrast enhancement and increased rCBV can be very specific indicators of underlying malignancy [35]. However, their performance in the timely identification of malignant lesions is not always sufficient. The majority of the enhancing lesions are high-grade, however, non-enhancing tumors may also be high-grade [9,10]. The incidence of non-enhancing malignant tumors has been statistically significant over the last two decades, thereby shaping the need for a more precise and sensitive imaging modality [10,36,37]. 

Our results confirm that the precise evaluation of accurate and reliable brain maps or tumor-metabolic changes requires close cooperation between an experienced nuclear medicine specialist (a neuro-nuclear specialist skilled in neurological imaging), a neuroradiologist and a neurosurgeon.

In conclusion, the diagnostic performance of static ^18^F-FET PET/CT provides the desired level of accuracy in establishing an early diagnosis in patients most likely to benefit from oncological treatment [38]. Our study confirmed the high sensitivity and high negative and positive predictive value of metabolic ^18^F-FET PET/CT, which also emphasizes its fundamental role during initial diagnosis. Comparing our results to the literature data, sensitivity was slightly higher (100% vs. 79–94%) whereas specificity was lower (40% vs. 57–72%) [15,39,40]. During our work, a mild association between SUVmax and cell proliferation was detected, which was in accordance with previously published results [41]. Nevertheless, TBRmax was not characteristic of either Ki-67 or IDH.

The results underlined the reliability of ^18^F-FET PET/CT and confirmed its vital role as an adjunct to MRI in the diagnostics of primary brain tumors. In addition to the therapeutic approach to malignant brain tumors, the evaluation of prognosis also requires a more detailed diagnostic tool [25,26,42].

Certain limitations in patient selection have to be addressed for comparing our results with existing literature data. Since only static ^18^F-FET PET scans were available, results were dichotomized into negative and positive cases. Thereby only positive scans can be unanimously translated into the clinical setting as malignant lesions. In addition, the low number of true ^18^F-FET negative cases presents a certain amount of distortion regarding the specificity of the examination. 

The study was limited to supratentorial tumors since histological sampling in brainstem lesions can be hazardous. Thus, in the absence of histological examination, accurate metabolic diagnostic build-up is more essential. 

Most scientific reports of recent years and subsequent guidelines present a more unequivocal recommendation about the use of amino acid PET/CT throughout the course of neuro-oncological therapy. Yet its indication for newly diagnosed tumor-suspect lesions is often lagging. 

## 5. Conclusions

As the main conclusion and current diagnostic recommendation, we support the ancillary use of ^18^F-FET PET measurements of all newly discovered non-enhancing intrinsic brain tumors. True biological tumor volume and the possible origin of the tumor are the most important factors both in surgical and oncological treatment planning. However, it is clear that the fusion and co-analysis of MRI and PET imaging is mandatory to obtain the best possible diagnostic results. Partly for this reason, further evaluation of which selected lesions should be considered for simultaneous PET/MR imaging and how this should be implemented in diagnostic guidelines is needed.

## Figures and Tables

**Figure 1 biomedicines-11-00128-f001:**
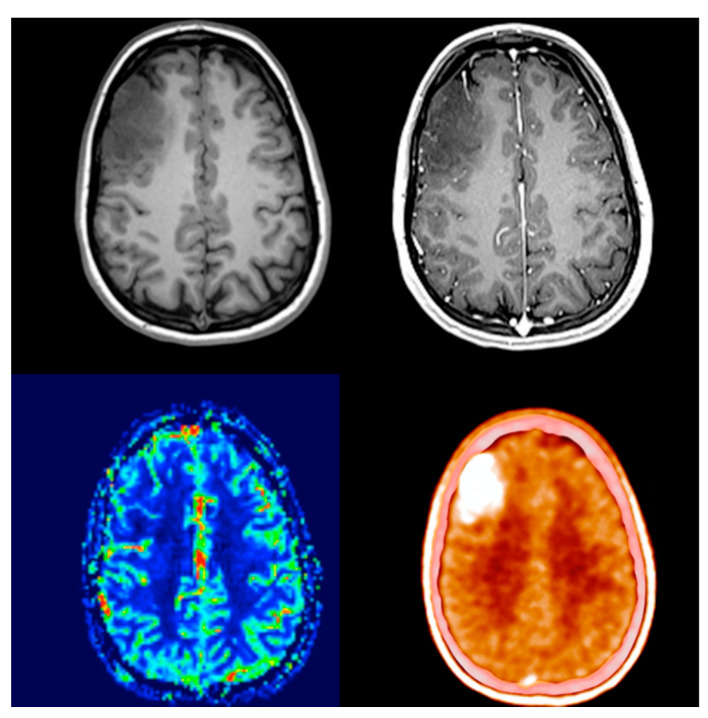
Thirty-two-year-old female with a right frontal non-enhancing mass known since the age of 14. Presenting for follow-up examination after 18 years of absence. The size of the lesion showed significant progression. Either contrast enhancement or elevated blood flow was present. PET/CT scan showed a smaller region and intense, well-circumscribed, nearly homogenous amino acid uptake (SUVmax: 4.95 vs. 1.57; TBRmax: 3.17). After surgical removal, histology revealed anaplastic oligodendroglioma with IDH mutation and 1p/19q co-deletion.

**Figure 2 biomedicines-11-00128-f002:**
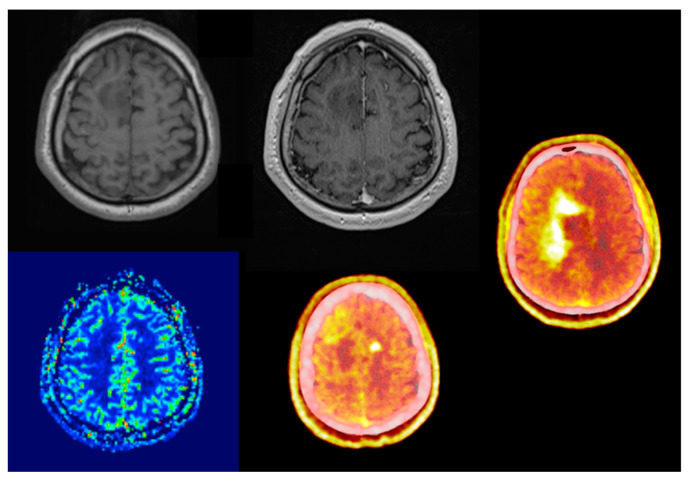
Sixty-three-year-old male, right frontal non-enhancing lesion known for over a year. During follow-up MRI scans no radiological progression was detectable. There were also no signs of elevated regional blood flow perfusion. No neurological deficit upon first and following presentations. PET/CT scan presented a butterfly-like very intense heterogeneous, lesion, significantly widened compared to the MRI lesion. Bilateral amino acid uptake (SUVmax: 55.81 vs. 1.58; TBRmax: 3.68) with a photopenic (SUV 1.4) area in the T1 hypointense area in the lateral part of the MRI lesion. The original T2 hyperintense lesion was photopenic (SUV: 1.4). Stereotactic biopsy revealed IDH-positive astrocytoma (grade 4), Ki-67 proliferation rate was 40%.

**Figure 3 biomedicines-11-00128-f003:**
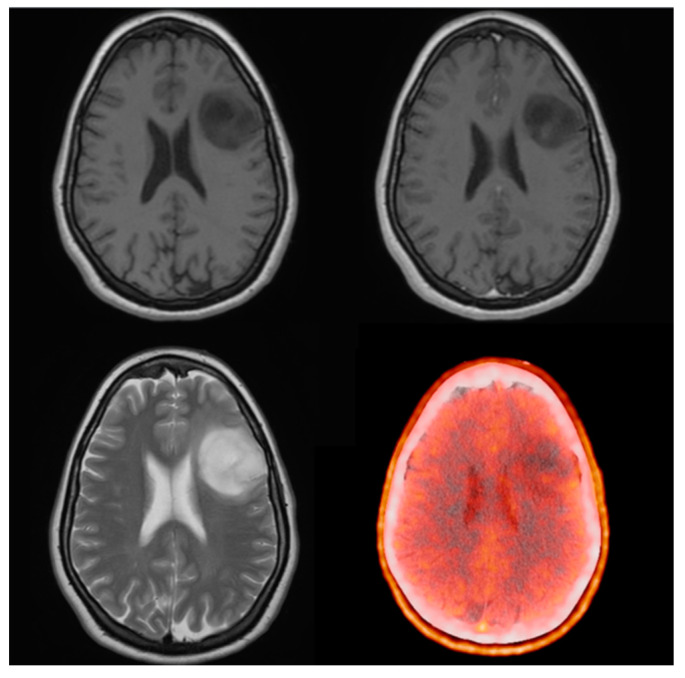
Thirty-one-year-old female, presenting with epileptic seizure. MRI showed a non-enhancing lesion in the left frontal operculum. On PET/CT the tumor was photopenic (SUVmax: 1.5 vs. 1.8, TBRmax: 0.8). Due to misinterpretation, the lesion was handled as low grade. First, a follow-up MRI made after 120 days already showed slight enhancement in the cranial part of the tumor. After waking surgical removal, histology revealed grade 4 astrocytoma with IDH mutation and a Ki-67 proliferation rate of 15%.

**Figure 4 biomedicines-11-00128-f004:**
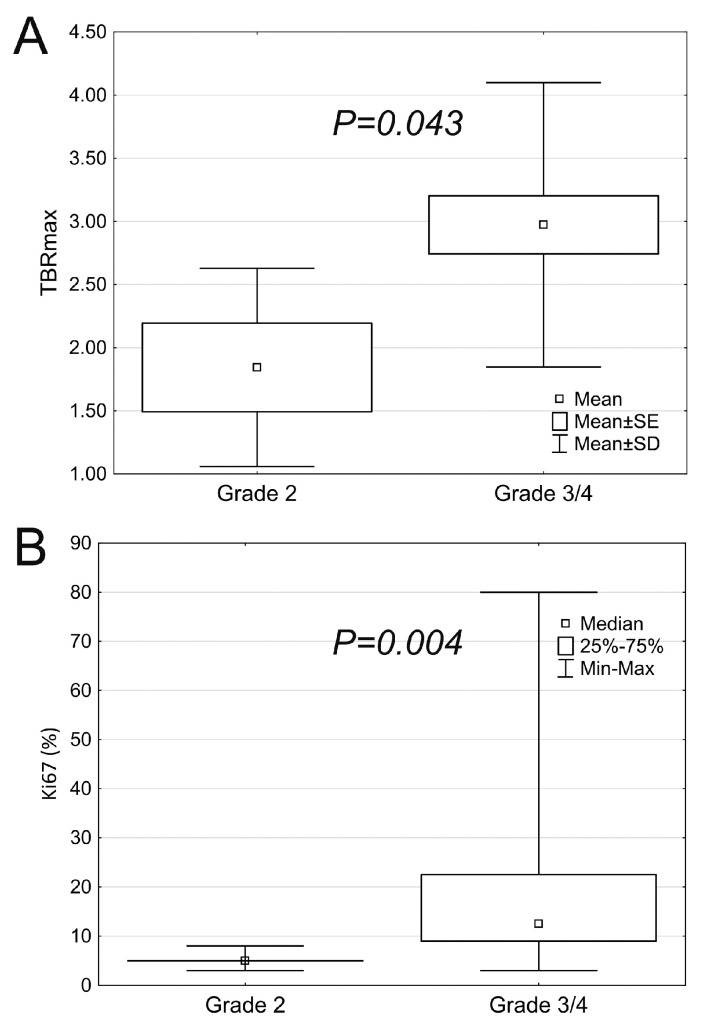
TBRmax (**A**) and Ki-67 (**B**) in low- and high-grade tumors.

**Figure 5 biomedicines-11-00128-f005:**
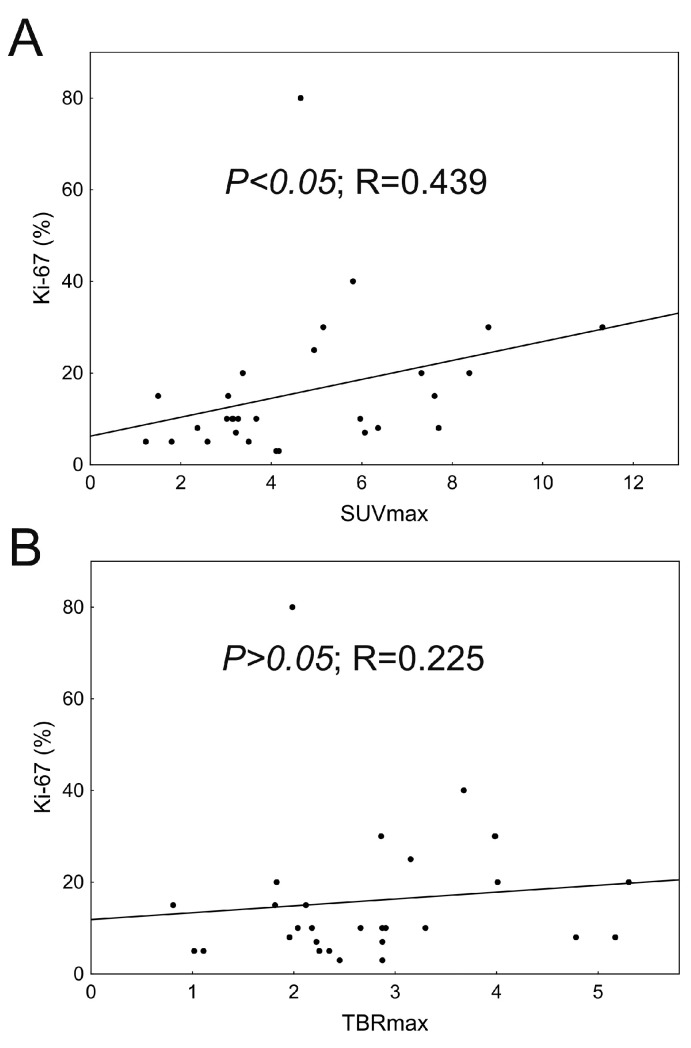
The association of SUVmax (**A**) and TBRmax (**B**)with Ki-67 (Spearman’s rank order correlation).

**Figure 6 biomedicines-11-00128-f006:**
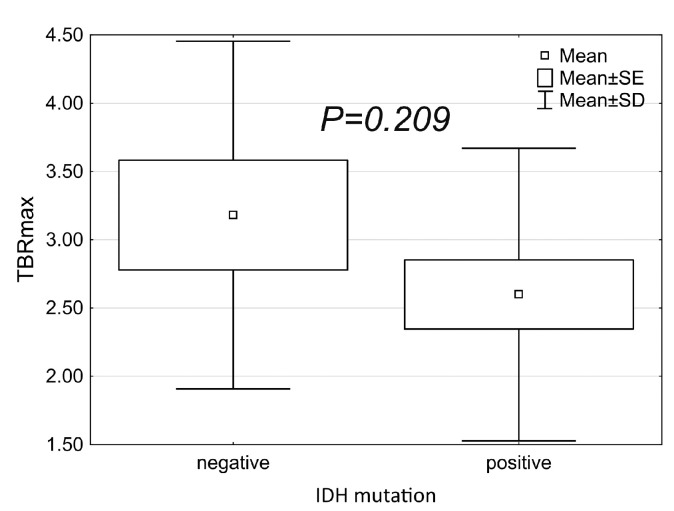
IDH mutation versus TBRmax.

**Table 1 biomedicines-11-00128-t001:** Characteristic of the studied brain tumor patients.

Characteristics		Cohort (*n* = 29)
Gender	Male	16 (55.3%)
	Female	13 (44.8%)
Age (year) (median; minimum-maximum)		42 (26–79)
Survival status	Alive	17 (58.6%)
	Dead	10 (34.5%)
	NA	2 (6.9%)
Survival (month) (median; minimum-maximum)		17 (2–108)
Laterality	Left	12 (41.4%)
	Right	17 (58.6%)
Localization	Frontal	18 (62.1%)
	Temporal	1 (3.4%)
	Occipital	1 (3.4%)
	Parietal	5 (17.2%)
	Insular	4 (13.8%)
Post ^18^F-FET histology	A2	2 (6.9%)
	A3	2 (6.9%)
	A4	7 (24.1%)
	GBM	6 (20.7%)
	E3	1 (3.3%)
	O2	3 (10.3%)
	O3	7 (24.1%)
	ETMR	1 (3.4%)
TBRmax (median; minimum-maximum)		2.66 (0.81–5.3)
TBRmax	<1.6	3 (10.3%)
	≥1.6	26 (89.7%)
Ki67 (%) (median; minimum-maximum)		10 (3–80)
Grade	2	5 (17.2%)
	3–4	24 (82.8%)
IDH-status	wild-type	10 (34.5%)
	mutant	18 (62.1%)
	NA	1 (3.4%)
MRI contrast enhancement	no	15 (51.7%)
	yes	14 (48.3%)
elevated MRI perfusion	no	17 (58.6%)
	yes	6 (20.7%)
	NA	6 (20.7%)

**Table 2 biomedicines-11-00128-t002:** Diagnostic performance of each modality at initial diagnosis regarding histological analysis.

	^18^F-FET PET	MRI Contrast	MRI Perfusion
Sensitivity	100%	52.9%	36.36%
	(16/16)	(9/17)	(4/11)
Specificity	40%	83.33%	100%
	(2/5)	(5/6)	(5/5)
Positive predictive value	84.21%	90%	100%
	(16/19)	(9/10)	(4/4)
Negative predictive value	100%	38.46%	41.67%
	(2/2)	(5/13)	(5/7)

**Table 3 biomedicines-11-00128-t003:** The association of tumor grade with the presence of contrast enhancement and elevated rCBV (Fisher’s exact test).

	Grade 2 (*n* = 5)	Grade 3/4 (*n* = 24)	*p*
Contrast enhancement			0.042
- no	5 (100%)	10 (58.3%)	
- yes	0 (0%)	14 (41.7%)	
Perfusion elevation			0.273
- no	5 (100%)	12 (50%)	
- yes	0 (0%)	6 (25%)	
- NA	0 (0%)	6 (25%)	

**Table 4 biomedicines-11-00128-t004:** The association of IDH mutational status with the evaluated contrast enhancement and perfusion elevation (Fisher’s exact test).

	IDH-Negative (n = 10)	IDH-Positive (n = 18)	*p*
Contrast enhancement			0.114
- no	3 (30%)	12 (66.7%)	
- yes	7 (70%)	6 (33.3%)	
Perfusion elevation			0.054
- no	3 (30%)	13 (72.2%)	
- yes	4 (40%)	2 (11.1%)	
- NA	3 (30%)	5 (27.8%)	

## Data Availability

All data used in the preparation of the paper are available on reasonable request.

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
