# Peer review of "Therapy Defining at Initial Diagnosis of Primary Brain Tumor—The Role of 18F-FET PET/CT and MRI"

_biomedicines, 2023, doi:10.3390/biomedicines11010128_

Round 1
Reviewer 1 Report
This is an interesting study reporting the diagnosis of primary brain tumors using 18F-FET PET/CT compared to MRI (based on contrast enhancement and perfusion). The paper is well-written and provides convincing evidence to support the role of 18F-FET PET/CT, particularly in the detection and diagnosis of IDH-mutant gliomas that do not show contrast enhancements on MRI. Some issues need to be clarified.
1. The cases that are included in the analysis are confusing. In Table 1, please add more information into 5 categories, the number of cases in: 1) the IDH wild-type vs. mutant; 2) the grade 2, 3, vs. 4; 3) TBRmax >= 1.6 vs. < 1.6; 4) MRI contrast enhancement (Yes vs. No); 5) elevated MRI perfusion (Yes vs. No). There were 30 patients in the survival status category- please correct this.
2. The results section was poorly organized, and difficult to understand. Please give sub-headings and clearly explain the PET and MRI diagnostic findings according to the actual case numbers that have interpretable PET and MRI results.
3. It was described that there were a total of 29 cases, but only 26 scans were 18F-FET PET positive. Did it mean that the other 3 cases were missed as false negatives, or did these 3 cases not have interpretable PET results? The writing is very unclear, so please clearly label the subsections to describe the PET false positive and false negative cases.
4. For MRI results, how were the contrast enhancement (Yes vs. NO), and elevated perfusion or rCBV (Yes vs. NO) determined? Was it based on the visual assessment or quantitative measurements (then, what is the threshold value)?
5. In Table 2, please give the case number in the numerator and the denominator (x/x), not just the percentage.
6. In Table 3, please add another category to show the IDH status, i.e. the case number with IDH wild-type vs. mutant or N/A.
Author Response
We do thank to Reviewer 1 for the valuable remarks. Please, find the applied modifications of our manuscript below. Please find the updated manuscript in the attachment.
- The cases that are included in the analysis are confusing. In Table 1, please add more information into 5 categories, the number of cases in: 1) the IDH wild-type vs. mutant; 2) the grade 2, 3, vs. 4; 3) TBRmax >= 1.6 vs. < 1.6; 4) MRI contrast enhancement (Yes vs. No); 5) elevated MRI perfusion (Yes vs. No). There were 30 patients in the survival status category- please correct this.
According to the recommendations we have added more details to Table 1 (e.g. IDH-status, grade, TBRmax, MRI contrast enhancement) providing a more exact description of the studied patient group. We have clarified our enrollments in Page 6, Row 189: since in one patient a non-tumorous lesion was confirmed by histologically later, we have excluded that case from further analysis.
- The results section was poorly organized, and difficult to understand. Please give sub-headings and clearly explain the PET and MRI diagnostic findings according to the actual case numbers that have interpretable PET and MRI results.
We updated the organization of the result section so that scan results are easier to follow and understand.
- It was described that there was a total of 29 cases, but only 26 scans were 18F-FET PET positive. Did it mean that the other 3 cases were missed as false negatives, or did these 3 cases not have interpretable PET results? The writing is very unclear, so please clearly label the subsections to describe the PET false positive and false negative cases.
We improved the description of the 3 FET PET negative scans and provided information about every negative scan as well in the results section (line 218-220).
- For MRI results, how were the contrast enhancement (Yes vs. NO), and elevated perfusion or rCBV (Yes vs. NO) determined? Was it based on the visual assessment or quantitative measurements (then, what is the threshold value)?
Contrast enhancement was evaluated by neuroradiologist specialist. Within the scope of this study it was not further analyzed yet than visual analysis, either present or not. Based on contrast enhanced images software calculates mean and relative cerebral blood volumes and this image is visually analyzed. White matter lesions are compared to healthy white matter lesion; cortical lesions are compared to either cortical or basal ganglia activity.
- In Table 2, please give the case number in the numerator and the denominator (x/x), not just the percentage.
Based on the recommendations, Table 2 has been updated, and the formula numerator and of denominator (x/x) has been added to it.
- In Table 3, please add another category to show the IDH status, i.e. the case number with IDH wild-type vs. mutant or N/A.
We examined IDH status separately in a different test. Since there was no grade II IDH negative patient, IDH status is only relevant in the grade 3/4 subgroup. On the other hand, as we have mentioned above, IDH-status was added to Table 1.

Reviewer 2 Report
The article " Therapy defining at initial diagnosis of primary brain tumor- the role of 18F-FET PET/CT and MRI " by Nagy et al. compares sensitivity, specificity, and predictive values of amino-acid-radiotracer-PET/CT with contrast-enhanced MRI and PWI in patients with primary brain tumors.
This is an interesting and valuable study, most parts of which appear well-written and suitable for publication.
However, as of yet, the manuscript needs clarification of methods in order to prove and understand results with less effort. Therefore, a final recommendation for the manuscript would require revision according to the following limitations:
In the current state, it is a bit difficult to make sense out of the sensitivity and specificity data mainly regaring MR-Imaging. To be able to better follow the methods and results, please clearly state in the methods section how MR-Imaging was classified as positive (possibly malignant?) or negative (possibly non-malignant?, if that is meant) and by which rational (e.g., interrater agreement on enhancement, .previous reports).
Additionally, I suggest to include the actual numbers leading to the percentages given in table 2 (e.g., 26/26 (100%) or similar). This would improve understanding and clarity for the reader.
There are some minor spelling errors, simply please check spelling throughout the report (e.g., line 103: “…Recent years…” è In recent years)
Abstract: Clear, no comments.
Introduction:
The last sentence of the first paragraph, appears grammatically incorrect. Please check.
Discussion:
In line 246-248 you write: “Meanwhile everyday experience shows growing incidence of non-enhancing malignant tumors thereby…...”. Please support your sentence with a reference and I suggest to leave out the passage “everyday experience shows”
Table 2: Please add a note to the caption of this table, that values represent comparisons with histological analyses
Author Response
We also would like to thank for the deep evaluations and helpful suggestions of Reviewer 2. Please, find the applied modifications of our manuscript below. Please find the revised manuscript in the attachment!
In the current state, it is a difficult to make a precise evaluation out of the sensitivity and specificity data mainly regarding MR-Imaging. To be able to better follow the methods and results, please clearly state in the methods section how MR-Imaging was classified as positive (possibly malignant?) or negative (possibly non-malignant?, if that is meant) and by which rational (e.g., interrater agreement on enhancement, previous reports).
The methods section has been updated with description for MRI evaluation (line 151-153). Signs of contrast enhancement and elevated perfusion were handled as possibly malignant lesions. Lack of aforementioned features were treated as cases that did not require usual oncological treatment.
Additionally, I suggest to include the actual numbers leading to the percentages given in table 2 (e.g., 26/26 (100%) or similar). This would improve understanding and clarity for the reader.
According to recommendation of Reviewer 2, the formula numerator and of denominator (x/x) has been added to Table 2, which was in parallel with Reviewer 1.
There are some minor spelling errors, simply please check spelling throughout the report (e.g., line 103: “…Recent years…” è in recent years)
The mentioned grammatical changes were performed.
Introduction:
The last sentence of the first paragraph, appears grammatically incorrect. Please check.
We have clearly checked and modified the questionable sentence.
Discussion:
In line 246-248 you write: “Meanwhile everyday experience shows growing incidence of non-enhancing malignant tumors thereby…...”. Please support your sentence with a reference and I suggest to leave out the passage “everyday experience shows”
We corrected the phrasing and updated the reference list with two articles about the incidence of non-enhancing malignant tumors.
Table 2: Please add a note to the caption of this table, that values represent comparisons with histological analyses
Thank you for the remark, we have added this relevant information to the caption.

Reviewer 3 Report
Title: Therapy defining at initial diagnosis of primary brain tumor-the role of 18F-FET PET/CT and MRI
Concise Summary:
Since the development of suitable amino acid radiotracers the role of PET/CT in the diagnosis in neurooncology is increasing. It has been postulated that amino acid PET/CT gives a relevant information in comparison of brain enhanced MRI. The authors reviewed the diagnostic performance of 18F-FET/PET CT over contrast enhanced and perfusion weighted MRI. Methodologically, the authors evaluate the sensitivity, specificity and negative and positive predictive values of metabolic 18F-FET PET/CT in respect of MRI according to the histopathology in the diagnosis of primary brain tumors. They conclude that 18F-FET PET/CT is more valuable than the enhancing MRI in the diagnosis of primary brain tumours.
Major criticisms:
Page 155. It is said that “Ki-67 (which refers mitotic activity).”. It is a mistake, because Ki67 is not referred to mitotic activity, but cell proliferation. In addition, which methodology was applied to measure the ki-67 immunostaining?
Page 184. It is indicated that “In one case the readings of 18F-FET data suggested non-tumorous lesion, which was also histologically confirmed.” Which was the histopathological diagnosis of these cases?
Page 193. Which was the histopathological diagnosis of three false positive cases?
Page 249. It is said that “…close cooperation between an experienced nuclear medicine specialist (a neuro-nuclear specialist skilled in neurological imaging), a neuro-radiologist and a neurosurgeon.” Being it truth, I consider that the pathologist (or neuropathologist) should be considered in this team. Could you explain because the neuropathologist is not considered in this respect?
Page 259. The authors state that “During our work, a mild association of SUVmax and mitotic activity was detected, which was in accordance with previously published results”. As it has been commented before, the mitotic activity has not been studied in this article. This maitake should be corrected.
Page 269. The authors comment that “Differentiating 18F- FET negative scans to non-tumorous or low-grade tumorous lesions were not possible.”- This sentence is confusing, because this issue is not studied in this work.
Title and Page 282. “… non enhancing intrinsic brain tumors.” However, 28 out 29 cases included in this study are gliomas. Which is the conclusion about the use of 18F- FET in embryonal tumours? In addition, could you explain in more detail the sentence “18F-FET/CT should always be suggested as therapy determining test”?
Minor criticism:
Title and text: “the role of 18F-FET PET/CT and MRI”. The title should avoid acronyms in order to render it comprehensible for readers. Then, O-(2-[18F]fluoroethyl)-L-tyrosine and magnetic resonance imaging should be used, respectively. In addition, the abbreviations list incorporated into the article is useful, but it does no avoid that the technical words should be introduced in the text previous word definition.
Page 59. “providing a more exact information about the dignity of the lesion.” Which is the meaning of the word “dignity” in this clinical context? Is it possible to say “dignity of the tumor”?
Figures 1-3. The legends should include a more detailed explanation of each one of the images.
Page 81. “Enabling adequate surgical intervention, correct sampling and subsequent oncological treatment.” This sentence is misspelled.
Pages 142-45 and figures 2-3.. It is indicated that “Histological analysis was performed in the same center …according to the current WHO CNS classification recommendation (4th and 5th editions).” Its is a mistake because the current classification is obviously fifth edition (WHO 2021). In this sense the diagnosis IDH mutated glioblastoma is not correct. These tumors should be diagnosed as IDH mutated astrocytomas.
Page 149. “Grade 2 histology was treated clinically as benign lesion.” This sentence is wrong since grade 2 diffuse type gliomas cannot be considered as benign lesions.
Page 173. Table 1. In "Post 18F-FET histology”. The histopathological diagnoses are referred to as acronyms. However, there are no meaning of these acronyms as a legend table In addition. the PNET type tumor should be indicated according to WHO 2021 brain tumor classification. Moreover, why the sum of patient survival status is thirty?
Conclusion:
The article addresses a relevant clinical issue through an interesting study of 18F-FET PET/CT and MRI and its relationship with patients’ survival. The author considerations about the issue of the article are adequate. However, I think that the authors should answer the reviewer observations before considering the article for publication.
Author Response
We would like to thank to Reviewer 3 for the accurate and valuable criticism. We have incorporated the suggestions as follows. Please find the revised manuscript in the attachment.
Major criticism
Page 155. It is said that “Ki-67 (which refers mitotic activity).”. It is a mistake, because Ki67 is not referred to mitotic activity, but cell proliferation. In addition, which methodology was applied to measure the ki-67 immunostaining?
The sentence has been corrected according to the recommendation of Reviewer 3. Ki-67 proliferation index was determined as part of a routine, fully automated (Leica Bond-Max) pathological protocol at the Department of Pathology and Experimental Cancer Research, Semmelweis University, Budapest, Hungary. The Ki-67 proliferation index (percentage of positive cells) was determined using the anti–Ki-67 antibody (Dako clone: MIB1 #M7240) by a neuropathologist expert (H.R., MD, PhD).
Page 184. It is indicated that “In one case the readings of 18F-FET data suggested non-tumorous lesion, which was also histologically confirmed.” Which was the histopathological diagnosis of these cases?
We have added this detail to the manuscript (in the recent version Row 190) that the diagnosis was demyelinating lesion, part of reactive tissue.
Page 193. Which was the histopathological diagnosis of three false positive cases?
As we described in the revisited version of the manuscript, those cases were all grade 2 gliomas. Based on the second opinions confirmed by the histopathological diagnosis, we also came to the conclusion that all these examinations and findings should be carried out in centers with special expertise.
Page 249. It is said that “…close cooperation between an experienced nuclear medicine specialist (a neuro-nuclear specialist skilled in neurological imaging), a neuro-radiologist and a neurosurgeon.” Being it truth, I consider that the pathologist (or neuropathologist) should be considered in this team. Could you explain because the neuropathologist is not considered in this respect?
We certainly agree with your opinion. All fields should be considered equally important during diagnostics and treatment, there is constant communication with the neuropathologist as well. Since the main scope of this review is the presurgical diagnostic we have not specifically emphasized the important role of the neuropathologist here.
Page 259. The authors state that “During our work, a mild association of SUVmax and mitotic activity was detected, which was in accordance with previously published results”. As it has been commented before, the mitotic activity has not been studied in this article. This mistake should be corrected.
We have replaced the incorrect term for “cell proliferation” in row 260 of the later version of the manuscript.
Page 269. The authors comment that “Differentiating 18F- FET negative scans to non-tumorous or low-grade tumorous lesions were not possible.”
This sentence is confusing, because this issue is not studied in this work.
The region of the morphological change visible on the MR images was always reflected on the PET images to the contralateral region, and strict care was taken to map the extent as accurately as possible. The pattern of FET-uptake in these cases was similar, compared to the opposite side, did not reach the significant literary "cut-off value".
Therefore, differentiating 18F-FET negative scans to non-tumorous or low-grade tumorous lesions were not possible due to the lack of dynamic measurements. The question in these cases if a well aimed biopsy carries any clinical benefit. Based on our experience so far, we believe that in these cases, the traditional, metabolic FDG-PET map can be used as ancillary examination for finding most relevant site of the tumor. Of course, only the neuropathologist can decide on the differentiation. But for clarification and simpler understanding we have chosen to delete this section.
Title and Page 282. “… non-enhancing intrinsic brain tumors.” However, 28 out 29 cases included in this study are gliomas. Which is the conclusion about the use of 18F- FET in embryonal tumours? In addition, could you explain in more detail the sentence “18F-FET/CT should always be suggested as therapy determining test”?
Thank you very much for the extremely interesting question, which we will pay close attention to in the future.As it is mentioned in the article and its references, there is a significant portion of non-enhancing tumours that need adjuvant oncological treatment. FET PET/CT proved to be very helpful identifying these patients. Since only one embryonal tumor was analyzed in our work we would not dare to state any conclusion about FET PET/CT application in these cases. However generally speaking malignant lesions have higher metabolism than non-neoplastic tissues. MRI scan was dubious so metabolic information was essential in order to decide about therapy.
Minor criticism
Title and text: “the role of 18F-FET PET/CT and MRI”. The title should avoid acronyms in order to render it comprehensible for readers. Then, O-(2-[18F]fluoroethyl)-L-tyrosine and magnetic resonance imaging should be used, respectively. In addition, the abbreviations list incorporated into the article is useful, but it does no avoid that the technical words should be introduced in the text previous word definition.
We thank you for your valuable comment and respect this criticism, but for practical reasons we think using the whole name of FET would be difficult in future literature searches.
Page 59. “providing a more exact information about the dignity of the lesion.” Which is the meaning of the word “dignity” in this clinical context? Is it possible to say “dignity of the tumor”?
Since the negative scans could also be non-tumorous we preferred using lesions instead of tumors. The context is meant for the dignity of the tumors of course. As it is a clinical study we rely on the oncological perspective.
Figures 1-3. The legends should include a more detailed explanation of each one of the images.
Thank you for this remark; however, we thought that extension of the explanation would not improve lucidity of the manuscript. We have applied these figures only for highlighting the questions arose of our work.
Page 81. “Enabling adequate surgical intervention, correct sampling and subsequent oncological treatment.” This sentence is misspelled.
The questionable phrase has been replaced to “This enables adequate surgical intervention, correct sampling and subsequent oncological treatment.”
Pages 142-45 and figures 2-3. It is indicated that “Histological analysis was performed in the same center …according to the current WHO CNS classification recommendation (4th and 5th editions).” It is a mistake because the current classification is obviously fifth edition (WHO 2021). In this sense the diagnosis IDH mutated glioblastoma is not correct. These tumors should be diagnosed as IDH mutated astrocytomas.
Thank you very much for underlying the importance of current guidelines. Some of these diagnoses were made prior 5th edition of WHO CNS tumor classification. In these particular cases we corrected the description and updated the patient characteristics according to the 5th edition.
Page 149. “Grade 2 histology was treated clinically as benign lesion.” This sentence is wrong since grade 2 diffuse type gliomas cannot be considered as benign lesions.
We based our differentiation on clinical and oncological perspective. Benign may not be the best description but since there is great chance they are treatable with surgery solely they have to be differentiated from grade 3 and 4 entities.
Page 173. Table 1. In "Post 18F-FET histology”. The histopathological diagnoses are referred to as acronyms. However, there are no meaning of these acronyms as a legend table in addition. the PNET type tumor should be indicated according to WHO 2021 brain tumor classification. Moreover, why the sum of patient survival status is thirty?
There was one non-neoplastic patient which were excluded from further statistical analysis as mentioned in the methods section (line 159-162); unfortunately, these values left from previous stage of the work – we have checked and corrected all the numbers. Again, some of the histological diagnoses were made prior to the 5th edition of WHO guide, but we updated the histological diagnosis based on the genetic status.
